# Antimicrobial Mechanism of Salt/Acid Solution on Microorganisms Isolated from Trimmed Young Coconut

**DOI:** 10.3390/microorganisms11040873

**Published:** 2023-03-29

**Authors:** Khemmapas Treesuwan, Wannee Jirapakkul, Sasitorn Tongchitpakdee, Vanee Chonhenchob, Warapa Mahakarnchanakul, Kullanart Tongkhao

**Affiliations:** 1Institute of Food Research and Product Development, Kasetsart University, Bangkok 10900, Thailand; 2Department of Food Science and Technology, Kasetsart University, Bangkok 10900, Thailand; 3Postharvest Technology Innovation Center, Science, Research and Innovation Promotion and Utilization Division, Office of the Ministry of Higher Education, Science, Research and Innovation, Bangkok 10400, Thailand; 4Department of Packaging and Materials Technology, Kasetsart University, Bangkok 10900, Thailand

**Keywords:** antimicrobial agents, coconut husk, microorganisms, mode of action, produce

## Abstract

This study investigated the inhibitory activity of organic solutions containing 5, 10, 15, 20 and 30% (*w/v*) sodium chloride and citric acid solution and 15:10, 15:15, 15:20 and 15:30% (*w/v*) sodium chloride (NaCl) combined with citric acid (CA) solution (salt/acid solution) for 10 min against microorganisms isolated from trimmed young coconut: *Bacillus cereus*, *B. subtilis*, *Staphylococcus aureus*, *S. epidermidis*, *Enterobacter aerogenes*, *Serratia marcescens*, *Candida tropicalis*, *Lodderromyces elongisporus*, *Aspergillus aculeatus* and *Penicillium citrinum*. Commercial antimicrobial agents such as potassium metabisulfite and sodium hypochlorite (NaOCl) were used as the controls. Results showed that 30% (*w/v*) NaCl solution displayed antimicrobial properties against all microorganisms, with s reduction range of 0.00–1.49 log CFU/mL. Treatment of 30% (*w/v*) CA solution inhibited all microorganisms in the reduction range of 1.50–8.43 log CFU/mL, while 15:20% (*w/v*) salt/acid solution was the minimum concentration that showed a similar antimicrobial effect with NaOCl and strong antimicrobial effect against Gram-negative bacteria. The mode of action of this solution against selected strains including *B. cereus*, *E. aerogenes* and *C. tropicalis* was also determined by scanning electron microscopy and transmission electron microscopy. *B. cereus and E. aerogenes* revealed degradation and detachment of the outer layer of the cell wall and cytoplasm membrane, while cytoplasmic inclusion in treated *C. tropicalis* cells changed to larger vacuoles and rough cell walls. The results suggested that a 15:20% (*w/v*) salt/acid solution could be used as an alternative antimicrobial agent to eliminate microorganisms on fresh produce.

## 1. Introduction

Most fresh produce is subjected to decontamination after harvest [1]. Chlorine and chlorine compound sanitizers are most commonly used by commercial processors due to their relatively low cost, easy use and high effectiveness against microorganisms [1,2,3]. However, negative issues associated with chlorine use, such as environmental impacts and worker health, have recently gained attention [1]. Sulfites, used commercially to maintain color, prolong shelf life and prevent microbial growth on fresh produce [4,5], have recently been banned in many countries due to possible allergic response symptoms in sulfite-sensitive people [6]. Chlorine and sulfites are still widely used but have negative effects on the environment and worker and consumer health [7,8]. Therefore, the use of alternative organic antimicrobial reagents such as sodium chloride [9], citric acid [10], or essential oils naturally occurring in wild species [11] have now attracted increased interest [9,12,13].

Sodium chloride (NaCl) has been used to flavor and preserve foods for thousands of years [2]. Salt is used as conventional sanitation in fig-processing plants as 5–10% (*w/v*) NaCl solution for 10 min [9]. Sodium and chloride ions associate with water molecules to decrease water activity [14,15], causing microbial cells to undergo loss of water from osmotic shock and inducing cell death or retarded growth [16].

Citric acid is a weak organic tribasic acid that exists as a white solid, is generally recognized as safe by the United States Food and Drug Administration (USFDA) [10,17] and is either naturally present in fruits and vegetables or synthesized by microorganisms through fermentation [18]. The antimicrobial activity of organic acids reduces the internal pH of microbial cells and disrupts substrate transport by altering cell membrane permeability [19].

Organic acids alter membrane permeability and accumulation of anions by reducing the pH value and inhibiting essential metabolic reactions [20]. They also damage microbial cells by interfering with the nutrient transport system through disruption of the cytoplasmic membrane, leading to cell leakage and interruptions in macromolecular synthesis [21].

Recently, a combination of salt and citric acid at 15% salt and 20% citric acid (salt/acid) solution was found to have a prominent effect on maintaining visual quality and retarding microbial growth on trimmed coconut surfaces [22,23].

This study investigated the inhibitory efficacy of salt/acid solution on microbial growth isolated from young coconut husk compared to commercial antimicrobial agents, and the mode of action of salt/acid solution on microbial structure change of selected resistance microorganisms.

## 2. Materials and Methods

### 2.1. Microbial Strains and Culture Conditions

Predominant microorganisms were isolated and detected in trimmed young coconut (TYC) by MALDI-TOF MS following [23]. Those microorganisms were used in this study and classified into four groups as (I) Gram-positive bacteria namely *Bacillus cereus*, *Bacillus subtilis*, *Staphylococcus aureus* and *Staphylococcus epidermidis*, (II) Gram-negative bacteria *Enterobacter aerogenes* and *Serratia marcescens*, (III) Yeasts as *Candida tropicalis* and *Lodderromyces elongisporus* and (IV) Molds as *Aspergillus aculeatus* and *Penicillium citrinum* [23].

### 2.2. Preparation of Microbial Inoculum

#### 2.2.1. Bacteria and Yeast Cell Preparation

A single inoculum of *B. cereus*, *B. subtilis*, *S. aureus*, *S. epidermidis*, *E. aerogenes* and *S. marcescens* was grown in trypticase soy broth (TSB; Merck, Darmstadt, Germany) at 35 °C, except for *B. cereus* which was incubated at 30 °C. Yeasts as *C. tropicalis* and *L. elongisporus* were grown in yeast malt medium broth at 30 °C. Cultures were maintained on slants at 4 °C and transferred monthly to maintain viability. A working culture was prepared by inoculating a loopful of culture into 10 mL of appropriate microbiological medium. The culture was then subjected to two successive transfers of 24 and 18 h before use.

#### 2.2.2. Fungal Spore Preparation

*A. aculeatus* and *P. citrinum* were grown on potato dextrose agar (PDA; Merck, Darmstadt, Germany) at 30 °C, transferred to malt extract broth (MEB; Merck, Darmstadt, Germany) and incubated for 7 days at 30 °C. An aliquot of 100 µL of each fungal inoculum was then pipetted onto a PDA plate and spread evenly over the surface. The plates were incubated at 30 °C for 7 days. Harvest was carried out by flooding the spores with 20 mL of sterile DI water with a sterile loop. The spore suspensions were filtered through two layers of sterile cheesecloth and then diluted with sterile 0.1% peptone water to obtain spore inoculum with the aid of a hemocytometer (Hauusser Scientific, Horsham, PA, USA) under a light microscope, as confirmed by colony counts in duplicate on PDA agar.

#### 2.2.3. Preparation of Selected Microbial Cocktail Inoculum

Significant microorganisms from our previous study with antimicrobial activities (Table 1) were selected for further investigation. *B. cereus* was selected because of its high resistance to all reagents, while *E. aerogenes* and *C. tropicalis* were selected due to their abundance on the coconut surface [23]. *C. tropicalis* was reported as a spoilage microorganism in coconut water [24]. Three strains of microorganisms, namely *B. cereus*, *E. aerogenes* and *C. tropicalis* were selected to prepare a microbial cocktail. They were cultivated following the method described in Section 2.2.1 and 10 mL cultures of each strain were transferred aseptically to 50 mL centrifuge tubes and vortexed for 10 s to ensure a homogenous cocktail (Vortex Mixer; Vortex-Genie-2 model G-560E, Radnor, PA, USA). The cocktail inoculum was centrifuged at 4500 rpm for 10 min (Centrifuge; Heraeus Biofugetrimo D-37520, Hanau, Germany). Each pellet was washed twice with sterile 0.85% NaCl and suspended in this solution to achieve a final concentration of 8 log CFU/mL of stock cocktail inoculum.

### 2.3. Antimicrobial Activity Assay

#### 2.3.1. Antimicrobial Agent Preparation

The antimicrobial properties of potassium metabisulfite (KMS), purchased from Chemipan (Bangkok, Thailand) Co., Ltd., and sodium hypochlorite (NaOCl) were investigated by diluting the appropriate amount of a commercial bleach solution (Haiter bleach, Chonburi, Thailand) containing 6% (*w/w*) NaOCl as available chlorine with water. The NaOCl solution was stored in a sterile screw-cap glass bottle and used within 2 h. Natural sodium chloride (NaCl) was purchased from a salt field in Samut Sakhon Province, Thailand and sterilized at 121 °C for 15 min before use. Citric acid anhydrous was purchased from COFCO Biochemical (Samut Sakhon, Thailand) Co., Ltd.

The concentrations of KMS were 1, 2.5, 3, 5 and 10% (*w/v*), while concentrations of NaOCl were 0.0005, 0.0010, 0.0015 and 0.002% (*v/v*). The concentrations of NaCl and citric acid were 5, 10, 15, 20 and 30% (*w/v*). The concentration ratios of salt/acid solution (SA) as sodium chloride and citric acid were 15:10, 15:15, 15:20 and 15:30% (*w/v*).

#### 2.3.2. Antimicrobial Activities of Single Inoculum

Antimicrobial activity of KMS, NaOCl, NaCl, citric acid and SA over time were determined against bacteria: *B. cereus*, *B. subtilis*, *S. aureus*, *S. epidermidis*, *E. aerogenes* and *S. marcescens*; yeasts: *C. tropicalis* and *L. elongisporus*; and fungal spores: *A. aculeatus* and *P. citrinum*. After 18 h of growth, the microorganisms and spores were serially diluted to 7 log CFU/mL. A volume of 10 mL was added in a 50 mL sterile centrifuge tube containing 1 mL of cell suspension and 9 mL of a mixture of either reagent or sterile water to obtain the desired concentration of antimicrobial agents. All centrifuge tubes were mixed by vortexing for 1 min and incubated at ambient temperature for 10 min to mimic the decontamination step of commercial processors. Survivors were monitored at intervals of 10 min by withdrawing 1 mL of sample, serially diluting and plating on PCA and PDA agar. Plates were incubated at optimal growth temperature for 24 to 48 h before enumeration.

#### 2.3.3. Antimicrobial Activities of Microbial Cocktail Inoculum on the Produce Matrix Model

The husk of green aromatic coconut was sliced using a sterile knife into 5 cm × 5 cm × 1 cm pieces and used as a produce matrix. These husk pieces were shaken in the cocktail suspension prepared according to Section 2.2.3 at 50 rpm by a shaker for 5 min to a final number of approximately 2–3 log CFU/25 cm^2^. The pieces of husk were then left to dry for 30 min. The inoculated coconut husk pieces were dipped in SA-15:20% (*w/v*) for 0 and 10 min at room temperature. Survivors were monitored by enumerating on MYP agar for *B. cereus*, MacConkey agar for *E. aerogenes* and PDA with 10% tartaric acid for *C. tropicalis*.

### 2.4. Investigation of Salt/Acid Solution Antimicrobial Mechanism

#### 2.4.1. Transmission Electron Microscopy

Suspensions of *B. cereus*, *E. aerogenes* and *C. tropicalis* were prepared according to Section 2.2.1 Each strain was centrifuged at 4500 rpm for 10 min (Centrifuge; Heraeus Biofugetrimo D-37520, Hanau, Germany). Each pellet was washed twice with sterile 0.1 M phosphate-buffered saline (PBS) pH 7.2 and suspended to the final solution concentration of 7 log CFU/mL stock inoculum. Salt/acid solution was then added to give a final concentration of 15% sodium chloride and 20% citric acid solution (*w/v*) and vortexed to ensure homogenous suspension (Vortex mixer; Vortex-Genie-2 model G-560E, Radnor, PA, USA). The pellets were fixed with 2.5% glutaraldehyde in 0.1 M PBS for 2 h at room temperature. The samples were postfixed with 1% *v/v* osmium tetroxide (OsO_4_) for 3 h and washed with the same buffer, dehydrated in a graded series of ethanol solutions (30, 50, 70, 80, 90 and 100% *v/v*) and then embedded in a Spurr low-viscosity embedding medium. Thin sections of the specimens were cut using a diamond knife on an Ultracut Ultramicrotome (Ultramicrotome Leica-Untracut; UCT, Wetzlar, Germany) and the sections were double-stained with uranyl acetate and lead citrate. The grids were examined with a HT7700 Transmission Electron Microscope (Hitachi, Tokyo, Japan) at an operating voltage of 80 kV. The control sample followed the same procedure without treatment with salt/acid solution.

#### 2.4.2. Scanning Electron Microscopy

A suspension of *B. cereus*, *E. aerogenes* and *C. tropicalis* was prepared and washed following Section 2.3 to prepare a stock inoculum. The husk of green aromatic coconut was sliced using a sterile knife into pieces 0.5 cm × 0.5 cm × 0.2 cm and used as a produce matrix model for microscopy. The pieces of husk were shaken in the suspension of *B. cereus*, *E. aerogenes* and *C. tropicalis* at 50 rpm for 5 min, and left to dry (under laminar airflow) for 10 min. The inoculated coconut husks were dipped in salt/acid solution at a ratio of 15:20% (*w/v*) for 10 min at ambient temperature. The treated samples were prepared by mounting on stubs in 2.5% glutaraldehyde in 0.1 M PBS for 18 h. Then, the specimens were rinsed twice with 0.1 M PBS and once with distilled water for 10 min. The specimens were dehydrated with a graded series of ethanol solutions (30, 50, 70, 95 and 100% *v/v*) for 15 min and changed three times at 100% *v/v*. Then, their surfaces were dried by a critical point dryer (Leica model EM CPD300, Vienna, Austria) and coated with gold (sputter coater, Balzers model SCD 040, Wetzlar, Germany). Electron micrographs were examined using a scanning electron microscope and energy-dispersive X-ray spectrometer (JEOL, JSM-IT-500HR, Tokyo, Japan). The control sample was performed following the same procedure without salt/acid solution treatment.

### 2.5. Statistical Analysis

All data were expressed as mean ± SD by measuring two independent replicates. Statistical analysis was carried out by analysis of variance (ANOVA) using SPSS program version 16.0. Duncan’s multiple range test (DMRT) was performed for post hoc multiple comparisons, and statistically significant differences were calculated at *p* < 0.05.

## 3. Results and Discussions

### 3.1. Antibacterial Activity Assay

#### 3.1.1. Microbial Inhibition of Individual Antimicrobial Agents

Activities of the commercial antimicrobial agents; sodium chloride solution, citric acid solution and salt/acid solution at 10 min are shown in Table 1. Initial populations of *Bacillus cereus*, *B. subtilis*, *Staphylococcus aureus*, *S. epidermidis*, *Enterobacter aerogenes*, *Serratia marcescens*, *Candida tropicalis*, *Lodderromyces elongisporus*, *Aspergillus aculeatus* and *Penicillium citrinum* ranged 6.3–8.4 log CFU/mL. Commercial control treatments of KMS solution at 1, 2.5, 3, 5 and 10% (*w/v*) for 10 min inhibited the growth of mold more than bacteria and yeast. Inhibitory activity on spore germination of *A. aculeatus* and *P. citrinum* was high at 6.30 and 6.44 log CFU/mL, respectively. Ref. [25] reported that sodium metabisulfite has strong antifungal activity. *B. cereus* treated with 10% (*w/v*) KMS solution for 10 min showed significant reduction compared with treatments of 1, 2.5, 3 and 5% (*w/v*). Results indicated that growth of *B. subtilis* and *S. epidermidis* were not significantly different and reduced after treatment with 1, 2.5, 3, 5 and 10% (*w/v*) KMS solution for 10 min. Conversely, treatment of 10% (*w/v*) KMS solution showed a reduction of 0.64–0.67 log on Gram-negative bacteria *E. aerogenes* and *S. marcescens* after treatment with all concentrations of KMS solution for 10 min. Inhibitory efficacies of all concentrations of KMS solution on *C. tropicalis* and *L. elongisporus* ranged 0.66–1.92 log CFU/mL. The maximum concentration of a 10% (*w/v*) KMS solution inhibited spore germination of mold as high as 6 log reduction, with a higher inhibitory effect on yeast and bacteria. The solution inhibited *C. tropicalis*, *S. epidermidis* and *B. cereus* at 5, 4 and 3 log reduction, respectively. However, other microorganisms showed less than 2 log reduction after treatment with KMS solution for 10 min as yeast *L. elongisporus* (1.92 log reduction), Gram-positive *B. subtilis* (1.89 log reduction), *S. aureus* (0.34 log reduction), Gram-negative *E. aerogenes* (0.67 log reduction) and *S. marcescens* (0.64 log reduction). Therefore, *B. cereus* and *L. elongisporus* were the most challenging for elimination by KMS solution, while molds were the most sensitive to KMS solution.

The inhibitory efficacy of NaOCl as a commercial control treatment was investigated on spoilage microorganisms identified from TYC at concentrations 0.0005, 0.0010, 0.0015 and 0.0020% (*v/v*) for 10 min. Results showed that NaOCl solution effectively inhibited growth of Gram-negative bacteria more than yeast and molds, with inhibitory effect on Gram-positive bacteria higher on *Staphylococcus* sp. than *Bacillus* sp. Treatment of NaOCl solution inhibited the growth of *B. cereus* at up to 3.04 log CFU/mL. Log reduction results of *B. subtilis* ranged 1.71–2.78 log CFU/mL, with an inhibitory effect of 0.0015% (*v/v*) NaOCl treatment for 10 min on *S. aureus*, *S. epidermidis*, *E. aerogenes*, *S. marcescens*, *C. tropicalis* and *L. elongisporus* as high as 7.82, 8.31, 8.43, 8.33, 6.44 and 6.44 log CFU/mL, respectively. Similar treatment on *A. aculeatus* and *P. citrinum* showed a low log reduction at 2.08 and 0.96 log CFU/mL, respectively. NaOCl solution was an effective antimicrobial agent at 0.0020% (*v/v*) on Gram-negative bacteria (approximately 8 log reduction), yeast (approximately 6 log reduction) and mold (approximately 3 log reduction), while Gram-positive bacteria showed log reduction on *Staphylococcus* sp. higher than 7–8 log CFU/mL but low inhibition on *Bacillus* sp. at 1–3 log CFU/mL reduction. *S. aureus*, *E. aerogenes*, *S. marcescens* and *L. elongisporus* showed high resistance to KMS solution but were inhibited by NaOCl at the minimum concentration of 0.0005% (*v/v*).

Inhibitory efficacies of microorganisms treated with NaCl at concentrations 5, 10, 15, 20 and 30% (*w/v*) for 10 min were observed. Results showed that NaCl solution at a high concentration of 30% (*w/v*) showed inhibitory efficacy at less than 1.02 log CFU/mL reduction. Ref. [9] also reported that using NaCl alone was less effective against mold *P. expansum*. In our study, NaCl solution at all concentrations had antimicrobial properties on *B. cereus* ranging 0.85–1.45 log CFU/mL reduction. Ref. [9] reported that using 4–12% NaCl alone reduced the growth of *B. cereus* on dried fig ranging 0.65–0.85 log CFU/g, while using 30% (*w/v*) NaCl solution did not inhibit the growth of *S. aureus* within 10 min. Conversely, NaCl solution at 30% (*w/v*) gave antimicrobial properties on *S. epidermidis*, *E. aerogenes*, *S. marcescens*, *C. tropicalis* and *L. elongisporus* at only 0.26, 0.51, 1.49, 0.76 and 0.46 log CFU/mL within 10 min, respectively. No inhibitory effect was recorded for NaCl solution on spore germination. Therefore, using the maximum concentration of NaCl solution as 30% (*w/v*) inhibited the growth of all strains of microorganism at less than 1.49 log reduction within 10 min.

The inhibitory effect of NaCl differed from the citric acid solution, and antimicrobial efficacy of 5, 10, 15, 20 and 30% (*w/v*) citric acid solution on microorganisms was investigated. Results indicated that citric acid solution at 30% (*w/v*) had stronger antimicrobial activity against Gram-negative bacteria than Gram-positive cocci, yeast, Gram-positive bacilli and mold. Ref. [26] reported that using 1% (*w/v*) citric acid treatment for 5 min reduced the growth of *Escherichia coli* O157:H7 at a 1.15 log reduction. In our study, using 5% (*w/v*) citric acid solution inhibited Gram-negative bacteria at 5–6 log CFU/mL reduction, while using the maximum treatment concentration of 30% (*w/v*) citric acid solution inhibited *B. cereus* and *B. subtilis* at 1.50 and 2.56 log CFU/mL, respectively with log reductions in *S. aureus*, *S. epidermidis*, *E. aerogenes* and *S. marcescens* high at 7.82, 8.73, 8.43 and 8.33 log CFU/mL. Antimicrobial testing of yeasts *C. tropicalis* and *L. elongisporus* gave 2.31 and 3.67 log CFU/mL, respectively (Table 1). Yeast inhibitory action with weak acid induces an energetic stress response that attempts to restore homeostasis and results in the reduction in available energy pools for growth and other essential metabolic functions [27].

By contrast, citric acid solution at 30% (*w/v*) did not inhibit spore germination of mold, *A. aculeatus* and *P. citrinum* within 10 min. Using citric acid solutions for 10 min as an antimicrobial agent inhibited Gram-negative bacteria and yeast at 8.43 and less than 3.67 log reduction, respectively. Citric acid solution at 30% (*w/v*) had an antimicrobial effect on Gram-positive bacteria similar to the commercial agent NaOCl, and inhibited *Staphylococcus* spp. and *Bacillus* sp. at more than 8.33 and 2.56 log reduction, respectively.

Our results showed that each antimicrobial agent had a different efficacy. A solution of NaOCl and citric acid was effective in inhibiting the growth of some bacteria and yeast but did not inhibit spore germination of mold, while treatment with KMS solution was more effective in inhibiting mold growth and some strains of bacteria and yeast within 10 min.

#### 3.1.2. Microbial Inhibition of Salt/Acid Solution

Antimicrobial activity of salt/acid NaCl and citric acid solution at ratios of 15:10, 15:15, 15:20 and 15:30% (*w/v*) on predominant microbial strains isolated from trimmed young coconut from our previous study [23] were investigated. Results indicated that using salt/acid solution as an antimicrobial agent showed a high microbial inhibitory effect on Gram-negative bacteria, Gram-positive cocci, yeast and Gram-positive bacilli. Results indicated that the salt/acid solution was more effective on *C. tropicalis* than the citric acid solution alone. Interestingly, the treatment of the salt/acid solution at a concentration of 15:10% (*w/v*) inhibited all *S. epidermidis* within 10 min, while the inhibitory efficacy of salt/acid solution at 15:20% (*w/v*) was more effective than using 20% (*w/v*) citric acid solution alone within 10 min. Treatment of salt/acid at 15:20% (*w/v*) for 10 min inhibited the growth of *E. aerogenes* and *S. marcescens* at 8.43 at 8.34 log CFU/mL, respectively. Though, excreted metabolites of *S. marcescens* were reported to have antimicrobial activities against Gram-negative bacteria and Gram-positive bacteria [28]. Ref. [18] found that the combination between 3% and 4% sodium chloride and 0.4% phytic acid against *E. coli* O157:H7 had a synergistic effect that increased microbial decontamination rate. Yeast inhibition with salt/acid solution at 15:20% (*w/v*) for 10 min gave log reduction of *C. tropicalis* and *L. elongisporus* at 4.19 and 1.86 log CFU/mL, respectively. Our results suggested that using a salt/acid solution showed a better inhibitory effect on yeast than using citric acid alone. Unfortunately, all concentrations of salt/acid solution used in our study did not inhibit mold growth.

#### 3.1.3. Comparison of Inhibitory Efficacy of Antimicrobial Agents

Antimicrobial efficacies of KMS, NaOCl, NaCl, citric acid and salt/acid solutions during the contact time of 10 min are shown in Table 2. Salt/acid solution had a stronger antibacterial effect against Gram-negative bacteria than Gram-positive bacteria and fungi. The higher resistance of Gram-positive bacteria was possibly related to their smoother surfaces, thicker peptidoglycan layer and lack of lipopolysaccharides [2,29]. A combination of NaCl and citric acid solution or salt/acid gave an antibacterial effect similar to NaOCl. The antibacterial and antifungal mechanism of salt/acid is not well understood but believed to occur from the interaction between turgor pressure in cells with a high concentration of NaCl [30], and disruption of substrate transport by altering cell membrane permeability [19], leading to the leakage of intracellular electrolytes and other constituents. Spores of *A. aculeatus* and *P. citrinum* were not inhibited by treating with salt/acid solution, while no antimicrobial agents showed comparable inhibitory effects against mold growth. The green hurdle approach, using salt/acid solution together with modified atmosphere packaging, was found to be as effective as KMS for inhibiting mold growth on trimmed young coconut during storage [23]. Thus, *B. cereus*, *E. aerogenes* and *C. tropicalis* were selected as a cocktail inoculum to investigate the antimicrobial mechanism of the salt/acid solution.

The optimal ratio of salt/acid solution on antimicrobial efficacy as 15% NaCl combined with 20% citric acid was selected for further testing the inhibition efficacy of microorganisms on the produce matrix model. The microbial structure change of *B. cereus*, *E. aerogenes* and *C. tropicalis* were also investigated using transmission electron microscopy (TEM) and scanning electron microscopy (SEM).

#### 3.1.4. Microbial Inhibition of the Salt/Acid Solution on the Produce Matrix Model

Figure 1 shows the reduction in a cocktail culture containing *B. cereus*, *E. aerogenes* and *C. tropicalis* on coconut husk treated with 15:20% (*w/v*) salt/acid solution for 10 min. Initial loads of each microbial strain were 4 log CFU/g. *E. aerogenes* was not detected on coconut husk treated with salt/acid solution. After treatment, *B. cereus* and *C. tropicalis* were detected at 1.94 and 0.11 log CFU/g, respectively, while *E. aerogenes* was not detected using the coconut husk model.

### 3.2. Salt/acid Solution Antimicrobial Mechanism

#### 3.2.1. Inhibitory Efficacy of Salt/Acid Solution on Microbial Structure by TEM

Bacteria and yeast were examined by transmission electron microscopy (TEM) to investigate the effect of salt/acid solution, as 15% sodium chloride combined with 20% citric acid (*w/v*) solution, on *B. cereus*, *E. aerogenes* and *C. tropicalis*. The TEM micrograph of *B. cereus* treated with salt/acid solution revealed alteration on the outer membrane, as shown in Figure 2. The non-treated cell showed an intact and apparent cell membrane as an electron-dense line (Figure 2A). Ref. [31] found that untreated *B. cereus* cells displayed smooth cell wall surfaces. Cells treated with the salt/acid solution showed altered outer membranes and some cytoplasm leaked out from the cell (Figure 2B). Disruption of the cell membrane was caused by high concentrations of sodium chloride and citric acid. The cells also displayed holes in the membranes and separation of the cell membrane and cell wall, similar to *Bacillus* spp. treated with 60 mg/mL of okra mucilage solution and 100 mg/L of catechins [31,32].

Non-treated cells of *E. aerogenes* showed a dense outer membrane with the inner layer separated by a low-density space (Figure 2C), while treated cells showed an altered outer membrane with an unclear and broken cell wall (Figure 2D). Results concurred with a report on membrane leakage of *Escherichia coli* after treatment with weak acids, such as acetic acid [33].

The TEM micrograph of *C. tropicalis* treated with salt/acid solution presented alteration on the outer membrane, as shown in Figure 2F. Treated cells showed the loss of intracellular material and a decrease in cytoplasmic density. Results concurred with [34], who found that cells of *C. tropicalis* treated with essential oils, thyme oil and crude clove extract exhibited evident damage. The control cell showed an intact, clear membrane as a dense line packed with cytoplasm (Figure 2E). Ref. [35] reported that organic acids such as citric acid caused alterations in membrane permeability and accumulation of anions, or a reduction in pH and inhibition of essential metabolic reactions [20]. The World Health Organization (WHO), (1998) explained that organic acid can damage microbial cells by interfering with the nutrient transport system and the disruption of cytoplasmic membranes, leading to cell leakage and interruptions in macromolecular synthesis [36].

#### 3.2.2. Inhibitory Efficacy of the Salt/Acid Solution on Microbial Structure by SEM

The three selected microbial strains *B. cereus*, *E. aerogenes* and *C. tropicalis* were inoculated on coconut husks before treatment with the salt/acid solution for 10 min. SEM results are shown in Figure 3, Figure 4 and Figure 5. Figure 3 shows the microstructures of *B. cereus* cells attached to coconut husk. The micrograph of the non-treated sample presented a regular and smooth surface (Figure 3A). Surfaces of the rod-shaped *B. cereus* remained unchanged, similar to a report by [31]. When the cells were treated with salt/acid solution for 10 min, cell lysis was observed (Figure 3B). Fissures and twisting were observed on treated cells similar to a report on *B. cereus* treated with okra mucilage solution [31]. Results were similar to the cells of *Bacillus* spp. that were twisted, helical and showed extensive shrinkage after treatment with medicinal plants and catechins [32,37]. This phenomenon concurred with the TEM results of treated *B. cereus* cells that showed damaged cell membranes. The SEM micrograph of *E. aerogenes* cells adhered to coconut husk is shown in Figure 4. The non-treated sample showed overall morphology of *E. aerogenes* cells with complete flagella and cells on the coconut husk (Figure 4A) similar to a report by [38].

After treatment with a salt/acid solution for 10 min, the micrograph showed that some of the flagella and cell membranes were damaged. The cell membrane exhibited a large number of dots, as shown in Figure 4B. These results were similar to damage on *E. aerogenes* cells in the presence of 5 mM furfural during fermentation. The surface of the bacterial cells had a large number of extracellular dots that appeared on the surface of the cell membrane [38]. Changes in the biophysical properties of microbial cells occurred because of osmotic stress generated from the solution concentration [2].

Morphologies of *C. tropicalis* on coconut husk as a non-treated sample and after treatment with salt/acid solution were investigated. Non-treated fungi showed ovoid yeasts with smooth and continuous surfaces similar to a report by [39]. The treated cells did not display intense cellular aggregation, surface cracks, scarce cells or large amounts of cellular debris, as reported by [40]. Therefore, results indicated that the SEM micrographs of the non-treated sample and treated sample were similar (Figure 5). The scanning electron micrograph results revealed that the salt/acid solution did not affect the cell surface and mainly affected intracellular materials, resulting in cytoplasmic density decrease, as revealed by the TEM result.

## 4. Conclusions

Antimicrobial activity of 15:20% (*w/v*) salt/acid solution for 10 min showed a stronger antibacterial effect against Gram-negative bacteria, Gram-positive bacteria, yeast and mold. Using this condition to inhibit microorganism growth was more effective than using sodium chloride and citric acid solution alone, with similar results to inhibition by NaOCl solution. Interestingly, no antimicrobial agents in this study showed an inhibitory effect against mold growth comparable to KMS. The mode of action of salt/acid solution (15% sodium chloride + 20% citric acid) against *B. cereus*, *E. aerogenes* and *C. tropicalis* was also determined by SEM and TEM. The microbial structure of treated cells with salt/acid solution for 10 min revealed degradation of the cell wall and detachment of the outer layer of the cell wall and cytoplasm membrane in bacteria, *B. cereus* and *E. aerogenes*, with cytoplasmic inclusion and rough cell wall in treated *C. tropicalis*. According to our report, this salt/acid solution could be used as an organic sanitizer for fresh-cut coconut industry. Nonetheless, further studies should be carried out to examine its proper concentration to be used in other fresh-cut fruit and vegetable industries.

## Figures and Tables

**Figure 1 microorganisms-11-00873-f001:**
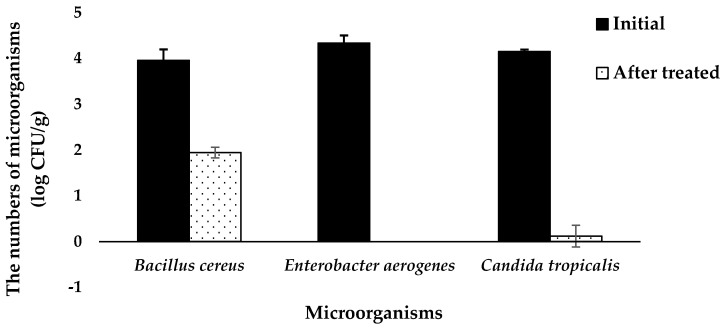
Numbers of microorganisms of *Bacillus cereus*, *Enterobacter aerogenes* and *Candida tropicalis* on coconut husk after treatment with 15% sodium chloride and 20% citric acid (salt/acid) solution for 10 min.

**Figure 2 microorganisms-11-00873-f002:**
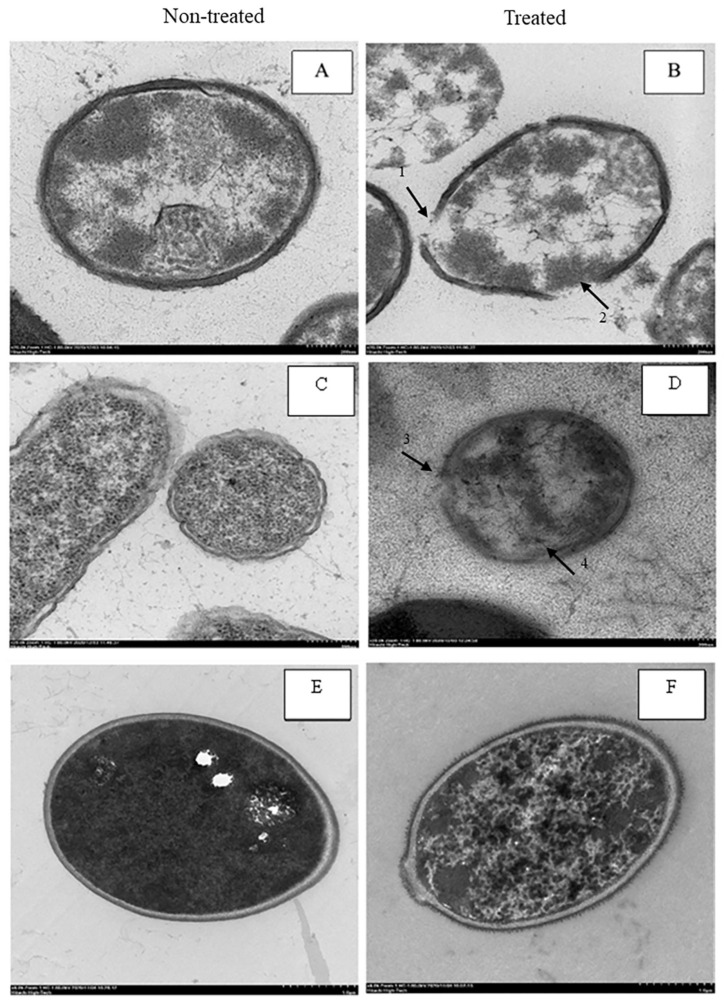
Transmission electron micrographs of non-treated *B. cereus* (**A**), *E. aerogenes* (**C**) and *C. tropicalis* (**E**) at 20,000×, 20,000× and 6000× magnification, respectively. The effect of 15:20% (*w/v*) salt/acid solution for 10 min on the microbial structure of *B. cereus* (**B**), *A. aerogenes* (**D**) and *C. tropicalis* (**F**) at 20,000×, 20,000× and 6000× magnification, respectively. Holes on cell surface (arrow #1, arrow #2 and arrow #3) and separation of cell membrane from cell wall (arrow #4).

**Figure 3 microorganisms-11-00873-f003:**
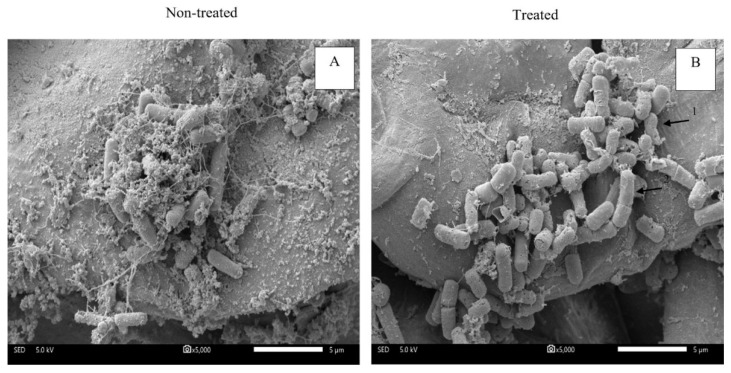
Scanning electron micrographs of *B. cereus* on coconut husk at 5000× magnification; non-treated (**A**) and treated with 15:20% (*w/v*) salt/acid solution for 10 min (**B**), respectively. Cell twisted (arrow #1) and cell elongated (arrow #2).

**Figure 4 microorganisms-11-00873-f004:**
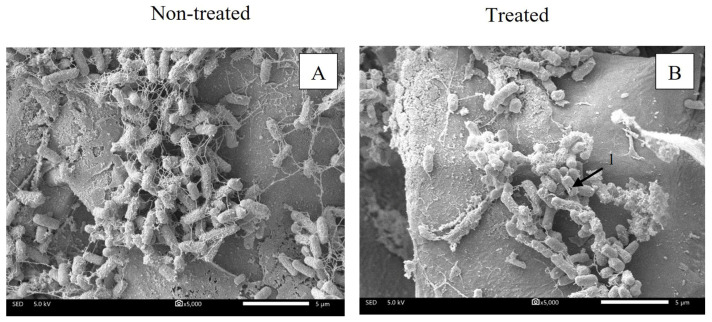
Scanning electron micrographs of *E. aerogenes* on coconut husk at 5000× magnification; non-treated (**A**) and treated with 15:20% (*w/v*) salt/acid solution for 10 min (**B**), respectively. Holes on cell surface (arrow #1).

**Figure 5 microorganisms-11-00873-f005:**
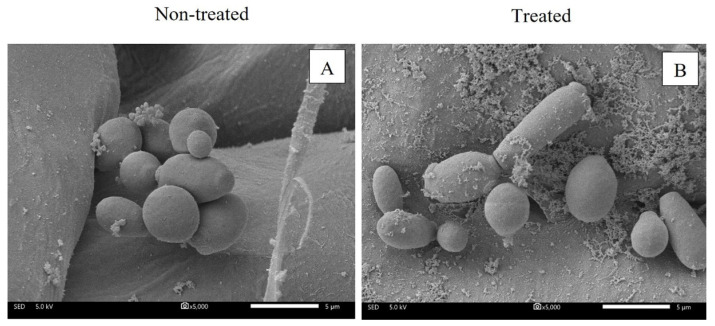
Scanning electron micrographs of *C. tropicalis* on coconut husk at 5000× magnification; non-treated (**A**) and treated with 15:20% (*w/v*) salt/acid solution for 10 min (**B**), respectively.

**Table 1 microorganisms-11-00873-t001:** Antimicrobial reagent concentrations on *Bacillus cereus*, *Bacillus subtilis*, *Staphylococcus aureus*, *Staphylococcus epidermidis*, *Enterobacter aerogenes*, *Serratia marcescens*, *Candida tropicalis*, *Lodderromyces elongisporus*, *Aspergillus aculeatus* and *Penicillium citrinum* during contact time of 10 min.

Reagents	Concentration (%)	Inhibitory Effect (Log Reduction ± SD)									
		(+ve) Bacteria				(−ve) Bacteria		Yeast		Mold	
		*B. cereus*	*B. subtilis*	*S. aureus*	*S. epidermidis*	*E. aerogenes*	*S. marcescens*	*C. tropicalis*	*L. elongisporus*	*A. aculeatus*	*P. citrinum*
KMS	1	1.30 ± 0.25 ^Cc^	1.97 ± 0.26 ^Ba^	0.53 ± 0.41 ^Da^	1.27 ± 0.09 ^Cc^	0.45 ± 0.14 ^Deb^	0.47 ± 0.14 ^DEabc^	0.34 ± 0.19 ^DEc^	0.66 ± 0.87 ^Da^	6.30 ± 0.35 ^Aa^	- ^* Eb^
	2.5	1.65 ± 0.32 ^Cb^	2.04 ± 0.02 ^Ca^	0.18 ± 0.14 ^Db^	1.86 ± 0.68 ^Cc^	0.68 ± 0.09 ^Da^	0.53 ± 0.23 ^Dab^	0.52 ± 0.13 ^Dc^	0.93 ± 0.92 ^Da^	6.30 ± 0.35 ^Ba^	6.44 ± 0.16 ^Aa^
	3	1.74 ± 0.19 ^Cb^	1.93 ± 0.22 ^Cb^	0.32 ± 0.12 ^Eab^	1.62 ± 0.05 ^Cbc^	0.60 ± 0.02 ^DEa^	0.34 ± 0.13 ^Ec^	0.80 ± 0.35 ^DEc^	0.94 ± 0.90 ^Da^	6.30 ± 0.35 ^Ba^	6.44 ± 0.16 ^Aa^
	5	1.74 ± 0.23 ^Db^	1.73 ± 0.06 ^Dab^	0.24 ± 0.13 ^Fb^	3.77 ± 0.30 ^Ca^	0.48 ± 0.10 ^Fb^	0.37 ± 0.01 ^Fbc^	3.80 ± 0.46 ^Cb^	1.21 ± 0.82 ^Ea^	6.30 ± 0.35 ^Ba^	6.44 ± 0.16 ^Aa^
	10	3.04 ± 0.19 ^Ea^	1.89 ± 0.00 ^Fab^	0.34 ± 0.06 ^Gab^	4.01 ± 0.04 ^Da^	0.67 ± 0.00 ^Ga^	0.64 ± 0.08 ^Ga^	5.37 ± 0.28 ^Cb^	1.92 ± 0.88 ^Fa^	6.30 ± 0.35 ^Ba^	6.44 ± 0.16 ^Aa^
NaOCl	0.0005	2.65 ± 0.32 ^Eb^	2.78 ± 0.21 ^Ea^	7.82 ± 0.09 ^Ba^	4.98 ± 0.14 ^Dc^	7.55 ± 0.07 ^Bb^	8.33 ± 0.05 ^Aa^	6.44 ± 0.16 ^Ca^	6.44 ± 0.88 ^Ca^	1.78 ± 0.26 ^Fb^	- ^Gc^
	0.0010	2.44 ± 0.09 ^Db^	1.88 ± 0.20 ^Eb^	7.82 ± 0.09 ^Ba^	6.81 ± 0.32 ^Cb^	7.63 ± 0.11 ^Bb^	8.33 ± 0.05 ^Aa^	6.44 ± 0.16 ^Ca^	6.44 ± 0.88 ^Ca^	1.26 ± 0.08 ^Fc^	0.22 ± 0.12 ^Gc^
	0.0015	2.65 ± 0.07 ^Db^	1.71 ± 0.12 ^Eb^	7.82 ± 0.09 ^Ba^	8.31 ± 0.04 ^Aa^	8.43 ± 0.03 ^Aa^	8.33 ± 0.05 ^Aa^	6.44 ± 0.16 ^Ca^	6.44 ± 0.88 ^Ca^	2.08 ± 0.24 ^Eab^	0.96 ± 0.33 ^Fb^
	0.0020	3.04 ± 0.19 ^Da^	1.75 ± 0.02 ^Fb^	7.82 ± 0.09 ^Ba^	8.31 ± 0.04 ^Aa^	8.43 ± 0.03 ^Aa^	8.33 ± 0.05 ^Aa^	6.44 ± 0.16 ^Ca^	6.44 ± 0.88 ^Ca^	2.26 ± 0.28 ^Ea^	1.92 ± 0.38 ^EFa^
NaCl	5	0.85 ± 0.09 ^Bc^	- * ^Cc^	- ^Ca^	0.01 ± 0.11 ^Cb^	0.65 ± 0.01 ^Bb^	1.46 ± 0.09 ^Ab^	0.24 ± 0.20 ^Cb^	0.56 ± 0.94 ^Bb^	- ^Ca^	- ^Ca^
	10	1.20 ± 0.14 ^Aab^	0.54 ± 0.40 ^Bb^	- ^Ca^	0.25 ± 0.12 ^BCab^	0.55 ± 0.01 ^Bc^	1.39 ± 0.11 ^Ab^	0.37 ± 0.14 ^BCab^	0.62 ± 0.87 ^Ba^	- ^Ca^	- ^Ca^
	15	1.23 ± 0.15 ^Aab^	- ^Dc^	- ^Da^	0.19 ± 0.03 ^CDab^	0.49 ± 0.05 ^BCd^	1.41 ± 0.04 ^Ab^	0.15 ± 0.03 ^CDb^	0.55 ± 0.94 ^Bb^	- ^Da^	- ^Da^
	20	1.45 ± 0.35 ^ABa^	1.08 ± 0.5 ^Ca^	- ^Da^	0.25 ± 0.02 ^Dab^	0.73 ± 0.07 ^Ca^	1.63 ± 0.11 ^Aa^	0.26 ± 0.23 ^Db^	0.64 ± 0.89 ^Cb^	- ^Da^	- ^Da^
	30	1.02 ± 0.22 ^Bbc^	0.58 ± 0.43 ^BCDb^	- ^Fa^	0.26 ± 0.31 ^DEFa^	0.51 ± 0.03 ^CDEcd^	1.49 ± 0.16 ^Ab^	0.76 ± 0.50 ^BCa^	0.46 ± 0.83 ^BCDb^	- ^Fa^	- ^Fa^
Citric acid	5	1.51 ± 0.08 ^Ea^	2.30 ± 0.00 ^Db^	0.98 ± 0.05 ^Fe^	8.33 ± 0.00 ^Aa^	6.14 ± 0.13 ^Bd^	5.24 ± 0.18 ^Cc^	0.33 ± 0.21 ^GHb^	0.50 ± 0.88 ^Gb^	- ^Ha^	- ^Ha^
	10	1.34 ± 0.08 ^Fa^	2.34 ± 0.00 ^Db^	1.60 ± 0.34 ^Ed^	8.33 ± 0.00 ^Aa^	6.54 ± 0.13 ^Bc^	5.47 ± 0.01 ^Cb^	0.88 ± 0.70 ^Gb^	0.52 ± 0.86 ^Gb^	- ^Ha^	- ^Ha^
	15	1.47 ± 0.10 ^Da^	2.46 ± 0.06 ^Ca^	2.73 ± 0.21 ^Cc^	8.33 ± 0.00 ^Aa^	6.65 ± 0.03 ^Bb^	8.33 ± 0.05 ^Aa^	0.92 ± 0.54 ^Db^	0.52 ± 0.91 ^Db^	- ^Fa^	- ^Fa^
	20	1.21 ± 0.08 ^CDa^	2.56 ± 0.13 ^Ca^	4.12 ± 0.16 ^Bb^	8.33 ± 0.00 ^Aa^	8.43 ± 0.03 ^Aa^	8.33 ± 0.05 ^Aa^	0.65 ± 0.72 ^Db^	1.01 ± 0.98 ^Db^	- ^Ea^	- ^Ea^
	30	1.50 ± 0.20 ^EFa^	2.56 ± 0.15 ^Da^	7.82 ± 0.09 ^Ba^	8.33 ± 0.00 ^Aa^	8.43 ± 0.03 ^Aa^	8.33 ± 0.05 ^Aa^	2.31 ± 0.11 ^Ea^	3.67 ± 0.75 ^Ca^	- ^Fa^	- ^Fa^
Salt/acid	15:10	0.86 ± 0.13 ^Ea^	3.50 ± 0.13 ^Ca^	4.22 ± 0.25 ^Bb^	8.31 ± 0.00 ^Aa^	8.43 ± 0.00 ^Aa^	8.34 ± 0.00 ^Aa^	3.07 ± 0.23 ^Db^	0.45 ± 0.22 ^Fc^	- ^Fa^	- ^Ga^
(NaCl: citric acid)	15:15	0.94 ± 0.37 ^Da^	3.30 ± 0.11 ^Cab^	4.80 ± 0.30 ^Bb^	8.31 ± 0.00 ^Aa^	8.43 ± 0.00 ^Aa^	8.34 ± 0.00 ^Aa^	3.43 ± 0.34 ^Cb^	0.66 ± 0.44 ^Dc^	- ^Ea^	- ^Ea^
	15:20	1.12 ± 0.13 ^Fa^	2.95 ± 0.45 ^Dbc^	6.67 ± 1.35 ^Ba^	8.31 ± 0.00 ^Aa^	8.43 ± 0.00 ^Aa^	8.34 ± 0.00 ^Aa^	4.19 ± 0.44 ^Ca^	1.86 ± 0.20 ^Eb^	- ^Ga^	- ^Ga^
	15:30	0.91 ± 0.13 ^Ea^	2.69 ± 0.35 ^Dc^	7.03 ± 0.92 ^Ba^	8.31 ± 0.00 ^Aa^	8.43 ± 0.00 ^Aa^	8.34 ± 0.00 ^Aa^	4.67 ± 0.31 ^Ca^	2.79 ± 0.17 ^Da^	- ^Fa^	- ^Fa^
Range of inhibitory effect			0–1.00	1.01–2.00	2.01–3.00	3.01–4.00	4.01–5.00	5.01–6.00	6.01–7.00	7.01–8.00	8.01–9.00

Values are average mean ± standard deviation of two replicates. Different superscript small letters within a column indicate significant differences between concentration at the level of *p* < 0.05. Different superscript capital letters within a row indicate significant differences between microbial species at the level of *p* < 0.05. Note: * = No inhibitory effect. Initial load of *B. cereus*, *B. subtilis*, *S. aureus*, *S. epidermidis*, *E. aerogenes*, *S. marcescens*, *C. tropicalis*, *L. elongisporus*, *A. aculeatus* and *P. citrinum* were 7.0, 7.2, 7.8, 8.3, 8.4, 8.3, 6.4, 6.4, 6.3 and 6.4 log CFU/mL, respectively.

**Table 2 microorganisms-11-00873-t002:** Efficacy of antimicrobial agents on *B. cereus*, *B. subtilis*, *S. aureus*, *S. epidermidis*, *E. aerogenes*, *S. marcescens*, *C. tropicalis*, *L. elongisporus*, *A. aculeatus* and *P. citrinum* during contact time of 10 min.

Microorganism	Minimum Concentrations with Maximum Inhibitory Effect of Antimicrobial Agents (% log Reduction)				
	KMS (% *w/v*)	NaOCl (% *v/v*)	NaCl (% *w/v*)	Citric Acid (% *w/v*)	Salt/Acid (% *w/v*)
Bacteria Gram-positive					
*B. cereus*	10 (99.9%)	0.0005 (99.77%)	10 (93.69%)	5 (96.69%)	15:20 (92.41%)
*B. subtilis*	10 (98.92%)	0.0005 (99.83%)	20 (91.68%)	5 (99.99%)	15:10 (99.96%)
*S. aureus*	10 (70.48%)	0.0005 (100%)	- *	30 (100%)	15:20 (99.9999%)
*S. epidermidis*	3 (94.62%)	0.0005 (99.998%)	10 (43.76%)	5 (100%)	15:10 (100%)
Bacteria Gram-negative					
*E. aerogenes*	10 (64.51%)	0.0005 (100%)	5 (77.61%)	5 (99.9999%)	15:10 (100%)
*S. marcescens*	10 (66.11%)	0.0005 (100%)	5 (96.53%)	5 (99.999%)	15:10 (100%)
Yeast					
*C. tropicalis*	5 (99.98%)	0.0005 (100%)	5 (42.45%)	30 (99.51%)	15:20 (9.91%)
*L. elongisporus*	10 (78.12%)	0.0005 (100%)	5 (72.45%)	30 (99.97%)	15:20 (98.61%)
Mold					
*A. aculeatus*	1 (99.999%)	0.0005 (98.34%)	-	-	-
*P. citrinum*	2.5 (99.999%)	0.0020 (39.74%)	-	-	-

Note: * = No inhibitory effect.

## Data Availability

The dataset generated during and/or analyzed during the current study are available from the corresponding author on request.

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
