# Peer review of "Antimicrobial Mechanism of Salt/Acid Solution on Microorganisms Isolated from Trimmed Young Coconut"

_microorganisms, 2023, doi:10.3390/microorganisms11040873_

Round 1

Reviewer 1 Report

Khemmapas Treesuwan and co-authors present a quality and well-written experimental manuscript describing antimicrobial mechanism of salt/acid solution on microorganisms isolated from trimmed young coconut.

Authors investigated the inhibitory activity of organic solutions containing 5, 10, 15, 20 and 30% (w/v) sodium chloride and citric acid solution and 15:10, 15:15, 15:20 and 15:30% (w/v) sodium chloride combined with citric acid solution (salt/acid solution) for 10 min against microorganisms isolated from trimmed young coconut: Bacillus cereus, B. subtilis, Staphylo- coccus aureus, S. epidermidis, Enterobacter aerogenes, Serratia marcescens, Candida tropicalis, Lodderromyces elongisporus, Aspergillus aculeatus and Penicillium citrinum. Commercial antimicrobial agents such as potassium metabisulfite and sodium hypochlorite were used as the controls. 

Authors found that 30% (w/v) NaCl solution displayed antimicrobial properties against all microorganisms, with reduction range 0.00-1.49 log CFU/mL. Treatment of 30% (w/v) CA solution inhibited all microorganisms in the reduction range of 1.50-8.43 log CFU/mL, while 15:20% (w/v) salt/acid solution was the minimum concentration that showed similar antimicrobial effect with NaOCl and strong antimicrobial effect against Gram-negative bacteria. Mode of action of this solution against selected strains including B. cereus, E. aerogenes and C. tropicalis was also determined by scanning electron microscopy and transmission electron microscopy. B. cereus and E. aerogenes revealed degradation and detachment of the outer layer of the cell wall and cytoplasm membrane, while cytoplasmic inclusion in treated C. tropicalis cells changed to larger vacuoles and rough cell walls. 

Finally, authors conclude that 15:20% (w/v) salt/acid solution could be used as an alternative antimicrobial agent to eliminate microorganisms on fresh produce.

==============================

Other comments:

1) Please check for typos throughout the manuscript.

2) With regards to Serratia marcescens microorganism – authors are kindly encouraged to cite the following article that describes specific enzymatic activity of S. marcescens. DOI: 10.3389/fphar.2018.00114

Reviewer 2 Report

The authors propose a manuscript titled “Antimicrobial mechanism of salt/acid solution on microorgan-isms isolated from trimmed young coconut”. The article is well structured. In particular, this study takes into consideration a topic aspect on the inhibitory activity of organic solutions containing in sodium chloride and citric acid solution, sodium chloride (NaCl) combined with citric acid (CA) solution against microorganisms (some Bacillus sp., Staphylo-coccus sp., Aspergillus aculeatus, Penicillium citrinum and others). Well known antimicrobial agents such as potassium metabisulfite and sodium hypochlorite were used as the controls. The results showed that NaCl solution displayed antimicrobial properties against all microorgan-isms, while thetTreatment of CA solution inhibited all microorganisms with a different the reduction range, and finally salt/acid solution was the minimum concentration that showed similar antimicrobial effect with NaOCl and strong antimicrobial effect against Gram-negative bacteria, suggesting that the salt/acid solution could be used as an alternative antimicrobial agent to eliminate microorganisms on fresh produce.

I appreciate the original idea of the work which with a few revisions will convince me and the editor to publish it on Journal.

1. Introduction

Well done and referenced. I have only few but crucial suggestions suggestions.

·        Most fresh produce is subjected to decontamination after harvest [choose references];

·        Chlorine and sulfites are still widely used but have negative effects on the environment and worker and consumer health [choose references];

·        Therefore, use of alternative organic antimicrobial reagents such as sodium chloride and citric acid [choose references], or those naturally occurring in wild species [Casella et al. 2023] have now attracted increased interest [choose references];

·        Sodium chloride (NaCl) has been used to flavor and preserve foods for thousands of years [choose references];

·        and is either naturally present in fruits and vegetables or synthesized by microorganisms through fermentation [choose references];

References to be added

·        Casella, F.; Vurro, M.; Valerio, F.; Perrino, E.V.; Mezzapesa, G.N.; Boari, A. Phytotoxic Effects of Essential Oils from Six Lamiaceae Species. Agronomy 2023, 13, 257. https://doi.org/10.3390/ agronomy13010257

·      

·     Arabidopsis thaliana (L.) Heynh. For botanical point of view is correct to cite the scientific name of the species (only for the first time in the manuscript) in the complete way.

2. Materials and Methods

·     Please when use for the first time the scientific name in the manuscript, write it also with the author that discovered it. Check whole document in this way.

ü B. cereus…

ü B. subtilis…

ü S. aureus…

ü S. epidermidis…

·     Table 1. Well done is clear but please standardize the font size;

3. Results and discussion

Weel done

·        Table 2. See my previous comment on Table 1;

Conclusion

·        Please spend tho more words in prospective of future research

References

·        Please check and format in the correct way.

Reviewer 3 Report

Minor revision

This article is  an importanjt contribution to find new anticontaminant to Most fresh produce .

10 microorganismes (bactrial reagent) were tested 

Results were suported by many analysis

I recommande for authors to add an abreviation list and add more recent work published in 2023 concerning this topic.

with regards
